# Characterising school-age health and function in rural Zimbabwe using the SAHARAN toolbox

Joe D. Piper[1,2]*, Clever Mazhanga[1], Gloria Mapako[1], Idah Mapurisa[1], Tsitsi Mashedze[1], Eunice Munyama[1], Marian Mwapaura[1], Dzivaidzo Chidhanguro[1], Eddington Mpofu[1], Batsirai Mutasa[1], Melissa J. Gladstone[3], Jonathan C. Wells[4], Lisa F. Langhaug[1], Naume V. Tavengwa[1], Robert Ntozini[1], Andrew J. Prendergast[1,2]

1 Zvitambo Institute for Maternal and Child Health Research, Harare, Zimbabwe, 2 Blizard Institute, Queen Mary University of London, London, United Kingdom, 3 Institute of Translational Medicine, University of Liverpool, Liverpool, United Kingdom, 4 Population Policy and Practice Research and Teaching Department, UCL Great Ormond Street Institute of Child Health, London, United Kingdom

* j.piper@qmul.ac.uk

## Abstract

### Introduction

We developed the School-Age Health, Activity, Resilience, Anthropometry and Neurocognitive (SAHARAN) toolbox to address the shortage of school-age assessment tools that combine growth, physical and cognitive function. Here we present i) development, acceptability and feasibility of the SAHARAN toolbox; ii) characteristics of a pilot cohort; and iii) associations between the domains measured in the cohort.

### Methods

Growth was measured with anthropometry, knee-heel length and skinfold thicknesses. Bioimpedance analysis measured lean mass index and phase angle. Cognition was assessed using the mental processing index, derived from the Kaufman Assessment Battery for Children version 2, a fine motor finger-tapping task, and School Achievement Test (SAT). Physical function combined grip strength, broad jump and the 20m shuttle-run test to produce a total physical score. A caregiver questionnaire was performed in parallel.

### Results

The SAHARAN toolbox was feasible to implement in rural Zimbabwe, and highly acceptable to children and caregivers following some minor modifications. Eighty children with mean (SD) age 7.6 (0.2) years had mean height-for-age (HAZ) and weight-for-age Z-scores (WAZ) of -0.63 (0.81) and -0.55 (0.85), respectively. Lean mass index and total skinfold thicknesses were related to WAZ and BMI Z-score, but not to HAZ. Total physical score was associated with unit rises in HAZ (1.29, 95% CI 0.75, 1.82, p<0.001), and lean mass index (0.50, 95% CI 0.16, 0.83, p = 0.004), but not skinfold thicknesses. The SAT was associated

**Data Availability Statement:** Data will be freely available as individual participant data with an accompanying data dictionary at http://ClinEpiDB. org from early 2022. This platform is charged with

ensuring that epidemiological studies are fully anonymised by removing all personal identifiers and obfuscating all dates per participant through application of a random number algorithm to comply with the ethical conduct of human subjects research. Researchers must agree to the policies and comply with the mechanism of ClinEpiDB to access data housed on this platform. Prior to that time, the data are housed on the ClinEpiDB platform at the Zvitambo Institute for Maternal and Child Health Research and available upon request from Ms. Virginia Sauramba (v. sauramba@zvitambo.com).

**Funding:** Wellcome Trust provided funding for this work and the salary of JDP. Grant number: 220671/Z/20/Z "Effect of early-life nutrition and WASH interventions on the long-term health of Zimbabwean children". AJP is funded by Wellcome Trust grant number 108065/Z/15/Z. Additional funding is from National Institutes of Health (NIH) Grant number R61HD103101, and research grants from the Thrasher Research Fund and Innovative Methods and Metrics for Agriculture and Nutrition Actions (IMMANA). The funders had no role in study design, data collection and analysis, decision to publish, or preparation of the manuscript.

**Competing interests:** The authors have declared that no competing interests exist

with unit increases in the mental processing index and child socioemotional score. The caregiver questionnaire identified high levels of adversity and food insecurity.

## Conclusions

The SAHARAN toolbox provided a feasible and acceptable holistic assessment of child growth and function in mid-childhood. We found clear associations between growth, height-adjusted lean mass and physical function, but not cognitive function. The SAHARAN toolbox could be deployed to characterise school-age growth, development and function elsewhere in sub-Saharan Africa.

## Introduction

There is an urgent need to reposition child health within the "survive, thrive and transform" global strategy [1]. This calls for a more holistic approach, recognising all stages of childhood as important for growth and development within a 'life-course' perspective. This framework recognises the negative exposures and positive opportunities that occur from conception to adulthood, with each life-stage building on previous stages [2]. It also highlights the interconnectedness and differences between growth, health, physical and cognitive function, allowing interventions to target multiple areas of development.

There has been remarkably little focus on school-age health outcomes between 5–14 years, particularly in low- and middle-income countries (LMICs), since routine health information systems do not capture information in this age group [3]. Mortality at age 5–14 years is disproportionately concentrated in LMICs, particularly sub-Saharan Africa [4]. Beyond survival, less is known about child growth and developmental trajectories after 5 years compared to younger ages, although recent studies have suggested that trends in height and BMI are highly variable in response to different social, nutritional and environmental factors, and are predictive of future health [5–9]. In particular, a recent study comparing global BMI trends showed that healthy school-age growth can either consolidate gains from early childhood or mitigate nutritional imbalances [9]. Conversely, in some countries (e.g. South Africa), children exhibit negative trajectories in health with a relative decline in height and an increase in BMI [9]. There is a pressing need to better understand the impact of school-age risk factors and protective factors, and the ability to mitigate early disadvantages to improve growth and development trajectories.

Refocusing attention on school-age health outcomes would increase our understanding of the timing of effective interventions to address growth, physical, cognitive, and socioemotional development in LMICs [10]. Following early-life nutrition interventions, some improvements in cardiovascular risk factors [11, 12], cognition [13] and later human capital [14] have been reported. Few studies have confirmed whether early growth gains translate into long-term improvements in both cognitive and physical function and, conversely, whether interventions showing little or no effect at early ages may nevertheless confer meaningful benefits on later measures of growth and function. For example, in the INCAP trial in 1970's Guatemala, children that received nutritional supplementation into early childhood had improved linear growth, which was associated with higher adult IQ scores, greater work capacity and earnings (among men) and greater schooling (among women) [14]. Since the INCAP trial, further portable techniques have been developed that can provide measurements of lean mass such as bioimpedance, enabling the differential contributions from body composition to be

investigated. Associations with fat and lean mass can also be explored with more recently developed measures of physical function such as handgrip strength, broad jump and shuttle run tests [15]. Similarly, associations with body composition and cognitive function can also be explored, as linear growth itself is a poor proxy for cognitive function [13]. Other key exposures such as years of schooling or the child's own reported socioemotional wellbeing may also impact cognitive function at school-age. By this age, it becomes easier to undertake more sensitive and specific assessments of cognitive development, compared to younger ages. Additionally, certain domains of brain function only begin to develop from 2 years of age, including expressive language and higher cognitive functions such as socio-emotional behaviour. To evaluate the effects of improved nutrition on all aspects of neurodevelopment therefore requires assessment at older ages. For example, follow-up of a trial of lipid-based nutritional supplements (LNS) delivered between 6–18 months in Ghana showed an effect on socio-emotional development at 4–6 years of age [16], which would not be measurable at 2 years. School-age assessments are more predictive of adult IQ and cognitive function than are early-life assessments, particularly as executive function can be better assessed [17]. Finally, school performance is itself a valuable cognitive outcome that can only be evaluated at older ages. Therefore school-age provides the opportunity to gain insights into growth reflecting body composition as well as detailed physical and cognitive function.

One reason for limited school-age data is a shortage of feasible and reliable tools combining measurements of growth, health, physical and cognitive function. Previous studies have individually shown that poor linear growth is associated with reduced cognitive development [18], grip strength [19] and increased cardio-metabolic risk factors [20]. However, the absence of holistic assessments combining neurodevelopment, physical fitness and growth has led to a call for further studies measuring a range of outcomes [13, 21]. Previous epidemiological work has identified height at 2 years as a strong predictor of later human capital [22]. However, assessing the efficacy of early-life interventions requires long-term follow-up to measure later functional outcomes. A comprehensive assessment is vital to understand school-age trajectories across different functional domains and the impact of risk and protective factors and potential interventions.

To overcome these gaps, we developed an integrated test battery combining measures of school-age growth, body composition, physical and cognitive function in sub-Saharan Africa, called the SAHARAN toolbox, specifically designed for use within low-resource communities. There were three objectives to this study: first, to develop the SAHARAN toolbox as a holistic measure of school-age growth, cognitive and physical function and demonstrate its feasibility and acceptability; second, to pilot it in a cohort of children in rural Zimbabwe to describe their growth and function; and third, to explore associations between key exposures and outcomes in this pilot cohort. The study adopted guidelines for describing pilot and feasibility studies [23], and the STROBE checklist for reporting of observational studies in epidemiology using the Equator network [24].

## Materials and methods

### 1. Development of the SAHARAN toolbox

For the first objective of developing a new toolbox, a conceptual framework was first developed to define the hypothesised relationships between a child's environment, growth, physical and cognitive function (Fig 1A). This conceptual framework provided a structure to guide how the tests would be selected and analysed. The underlying environmental factors were based on previous work by Prado [25, 26]. We included assessments of fat and lean mass to evaluate quality of growth, and their relationship with physical and cognitive function. The School-Age Health,

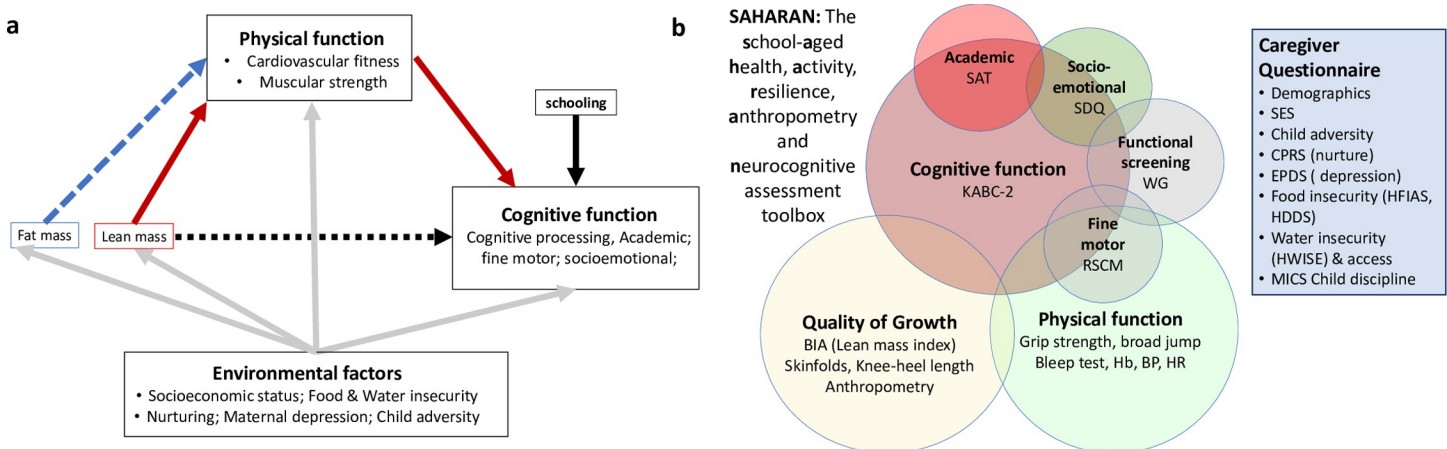

**Fig 1. (a).** Conceptual framework to inform the choice and design of tools. **(b).** The SAHARAN toolbox. A 4-hour assessment comprising child growth, body composition, cognitive and physical function assessment with simultaneous caregiver questionnaire. **KABC-II:** Kaufman Assessment Battery for Children 2nd edition, SDQ: Strength and difficulties Questionnaire, WG: Washington Group / UNICEF Child Function module, RCSM: Rapid continuous sequential movements, BIA: Bioimpedance analysis, Hb: Haemoglobin, BP: Blood pressure, HR Heart rate, SES: Socioeconomic status, CPRS: Child-parent relationship scale, EPDS: Edinburgh postnatal depression score, HFIAS: Household Food Insecurity Assessment Scale, HDDS: Household Dietary Diversity scale, HWISE: Household Water Insecurity Experiences Scale, MICS: Multi-indicator cluster survey (UNICEF) for Discipline Questionnaire.

Activity, Resilience, Anthropometry and Neurocognitive (SAHARAN) toolbox (Fig 1B) was designed to test these relationships, using a stepwise approach described in detail in S1 File [27].

In brief, we first conducted a detailed literature review, which identified recent systematic reviews describing tools for measuring cognitive function [13, 28, 29], a recent toolkit published by the World Bank [30], and a test battery for physical function [15, 31]. Second, individual assessment tools were screened for their use in LMICs [32, 33] using selection criteria based on the COSMIN tool (see S1 File) [34]. Third, a range of international child development, nutrition and sports science experts were contacted to provide input on each tool, including their content validity and applicability for a rural sub-Saharan African context [35]. Fourth, pre-testing and cognitive interviewing instruments led to further adaptation and selection of tests for the final toolbox (Fig 1B). Feasibility and acceptability were evaluated through face validity, and feedback from data collectors, caregivers and children performing the tests.

Detailed descriptions of the final test battery are provided in (S1 to S4 Tables in S1 File). Briefly, the SAHARAN toolbox comprises a single assessment conducted with the child and caregiver, focused on three domains: growth and body composition, physical function and cognitive function (Fig 1B). Growth is assessed using standard anthropometry (weight using portable scales (Seca, Germany), height using a Shorrboard (Weigh and Measure, Olney, USA), head, arm, waist, hip and calf circumferences using measuring tapes (Weigh and Measure, Olney, USA). Body composition is assessed using three techniques. First, bioimpedance analysis (BIA) estimates lean mass, with the child recumbent using the Bodystat 1500 MDD instrument (BodyStat, Isle of Man, UK) [36]. Second, skinfold thicknesses (SFT) measure peripheral (calf and triceps) and central (subscapular) subcutaneous fat using a skinfold caliper (Holtain, Crosswell, UK). Third, knee-heel length examines relative body proportions [37] using a commercial knemometer (Weigh and Measures, Olney, USA). For physical function three physical tests were included. First, handgrip strength is measured using a dynamometer (Takei, Japan) held vertically downwards [19] with the elbow extended to provide the most appropriate positioning [38]. The child stands and squeezes as hard as they can for up to five seconds, repeating this three times after a suitable break. The maximum value is used for analysis for each hand. Second, broad jump [19] measures leg strength. The child stands behind a

line marked on the ground, bends the knees, swings the arms and has a two-footed take-off and landing. Third, the shuttle-run test (SRT) assesses cardiorespiratory function [39]. The child runs repeatedly between two markers that are 20 meters apart; timed 'beeps' are provided using a Bluetooth speaker connected to a tablet using the "Beep test" free android app (Beep Test, Ruval Enterprises, Canada). Measurements of blood pressure, heart rate and haemoglobin were also included in the assessment battery. Haemoglobin is measured using a fingerprick point-of-care test (Hb301, Hemocue, Sweden), and blood pressure using an electronic or manual sphygmomanometer (Medisave, UK).

Cognitive function is measured using the mental processing index (MPI), calculated from 8 core subtests of the Kaufman Assessment Battery for Children, 2nd edition (KABC-II) [33, 40]. Academic function is assessed by the school achievement test (SAT) which measures reading, writing and numeracy. Fine motor function is evaluated by the time to complete sequential finger tapping [41], with a shorter time representing improved fine motor function. Socioemotional function is measured by the caregiver-completed Strengths and Difficulties Questionnaire (SDQ) which is free to download online (https://www.sdqinfo.com/) [42]. The number of years of schooling completed by the child is reported by the caregiver, as a key exposure for cognitive function. A short, confidential child questionnaire asks about their perceived level of socioemotional support, which has also been shown to be a key exposure for cognitive function [43]. Child disability and overall caregiver-reported function are measured using the Washington Group / UNICEF Child Functioning Module [44]. A detailed caregiver questionnaire measures household demographics, adversities [45, 46], socioeconomic status [47], food security [48, 49], water security [50], and caregiver nurturing [51], depression [52] and discipline [53, 54]. Wherever possible, pictorial Likert scales are used to assist the caregiver with choosing a response.

## 2. Pilot cohort to evaluate the SAHARAN toolbox

Our second objective was to deploy the SAHARAN toolbox in a cross-sectional pilot cohort of children aged 7 years, whose characteristics are presented here. The pilot study was conducted in Zvamabande (rural) and Makusha (urban) regions of Shurugwi district in Midlands Province, Zimbabwe. During sensitisation events, the proposed tests were explained to communities. Eligible children aged 7 years were identified by community health workers (CHW), then 80 children were randomly selected by computer, and a sensitisation visit to the family was conducted by the CHW. If the family expressed interest in participating, a date was agreed for a visit.

Assessments were undertaken in the community using portable equipment (S1 Fig in S1 File). A handwashing station was erected and facemasks distributed, in line with district COVID-19 policies. One or two tents were pitched close to the household with four folding chairs, where the mother and child could see each other at all times. The child cognition measurements were administered in the tent using a folding table. Data were collected using Open Data Kit (ODK) [55] on tablet computers (Samsung Galaxy Tab A) for most measurements, enabling appropriate data skips and plausibility checks; some cognition measurements used paper forms (KABC-II, School Achievement Test). Detailed standardisation exercises were conducted to assess intra- and inter-rater reliability as part of the development of the toolbox.

## 3. Exploring associations within the pilot cohort

Our third objective was to explore the associations between key exposures and outcomes using components of the SAHARAN toolbox. For physical function, our exposure of interest was growth, and outcomes were individual measures of handgrip strength (Kg), broad jump distance

(cm) and level achieved in the shuttle-run test. Total physical score was calculated by adding the standardised scores from the three physical function tests. For cognitive function, our exposures of interest were growth, years of schooling, and the child's perceived socioemotional support. Our outcomes were the KABC-II total score (the MPI), total score achieved in the SAT, fastest time to complete sequential finger tapping, and total difficulties score on the SDQ.

## 4. Participant and public involvement

School-age participants and their caregivers were not involved in setting the research question or measures. However, they were involved in design, development and implementation of the questionnaires and tools. All tests to measure school-age child growth, physical and cognitive function were discussed with the District Health Executive, community leaders, and health centre committees in Zvambande and Makusha. All questionnaires underwent cognitive interviewing with community members who suggested alterations and feedback. Tools were then pre-tested with local families to ensure acceptability, with feedback sought from children and caregivers. Extensive sensitisation of the communities was undertaken by the study team and community health workers in conjunction with screenings of a community-made film which explored their previous experiences of being involved in research. During sensitisation and consent visits, it was explained that findings from the current study would not be shared because the aim was to assess the feasibility and acceptability of the SAHARAN toolbox. Hence, as part of consent, it was clarified to participants that this was a single visit with no further follow-up.

## 5. Data analysis

First, we defined the specific growth measures from our test battery to use as exposure variables, by performing a Pearson correlation analysis using all growth variables (S6 Table in S1 File). Growth measurements were converted to Z-scores using WHO reference standards [56]. BIA data were converted into lean mass index, defined as $1/Z$ (the average impedance), which is independent of height [57]. The impedance index, defined as height-squared divided by average impedance ($H^2/Z$), was also calculated as a direct estimate of relative lean mass; this incorporates height into the estimation, as lean mass always scales strongly with height [57]. Impedance index therefore acts as a composite marker of muscle and organ mass relative to height. Phase angle was used as a marker both of cell mass and tissue health [58, 59]. Variables with a Pearson's correlation coefficient >0.79 indicated a strong co-linear relationship [60] and so were not included in further analysis (BMI Z-score, knee-heel length, hip circumference, impedance index and individual skinfold thicknesses). Hence the key growth variables selected as exposure measures were: HAZ, WAZ, head circumference, MUAC, waist circumference, calf circumference, lean mass index (LMI), phase angle, total skinfold thickness and haemoglobin.

Second, construct validity and internal consistency of the new test battery were assessed by examining the relationships between different tests within similar physical and cognitive function domains (convergent validity) using least squares regression. We therefore explored the association between each of the three main physical function variables, since two represent measures of strength (grip strength and broad jump) and one represents cardiovascular fitness (shuttle-run test); S7 Table and S2 Fig in S1 File. We also combined the standardised physical function tests to make a total physical score (TPS). We similarly explored the association between each of the main cognitive measures (S8 Table and S2 Fig in S1 File) to examine internal consistency, since each measure represents different domains of cognitive function. Since

**Table 1. Baseline characteristics of participants, household and selected adversity factors in the sample.**

| Child characteristics | All (N = 80) |
|---|---|
| Female, N (%) | 39 (49%) |
| Age, yrs; mean (SD) | 7.6 (0.2) |
| HAZ; mean (SD) | -0.63 (0.8) |
| Stunted (HAZ<-2), N (%) | 2 (3%) |
| WAZ; mean (SD) | -0.55 (0.8) |
| Underweight (WAZ <-2), N (%) | 3 (4%) |
| BMI-Z score, mean (SD) | -0.27 (0.9) |
| Enrolled in school, N (%) | 15 (19%) |
| Reported schooling at home, N (%) | 33 (41%) |
| Years in school; mean (SD) | 3.1 (0.7) |
| **Caregiver characteristics** | **All (N = 80)** |
| Mother; N (%) | 54 (68%) |
| Grandmother; N (%) | 20 (25%) |
| Other; N (%) | 6 (7%) |
| Age, years; mean (SD) | 39.8 (11.7) |
| Years of schooling; mean (SD) | 9.1 (2.7) |
| Married, N (%) | 63 (79%) |
| EPDS score; mean (SD) | 6 (5) |
| **Household characteristics** | **n** |
| Household size; median (IQR) | 3 (2,5) |
| Electricity in home; N (%) | 9 (11%) |
| Drinking Water > 5 mins walk; N (%) | 62 (78%) |
| Own radio; N (%) | 59 (74%) |
| Own phone; N (%) | 79 (99%) |
| Own TV; N (%) | 36 (45%) |
| Own car/truck; N (%) | 8 (10%) |
| Own solar panel; N (%) | 62 (78%) |
| Own 2 or more children's books; N (%) | 29 (36%) |
| HFIAS median score (IQR) | 6 (1, 10) |
| HDDS median score (IQR) | 8.5 (7,10) |
| HWISE median score (IQR) | 0 (0,1) |
| **Adversities since child's birth** | **All (N = 80)** |
| Death in household | 26 (33%) |
| Adult sick / injured > 3 mo | 11 (14%) |
| 1 crop failure | 60 (75%) |
| 2 or more crop failures | 37 (46%) |
| Loss of family possessions | 27 (34%) |
| Alcohol problem | 16 (20%) |
| Debt problem | 16 (20%) |
| >2 Child hospital admissions | 2 (3%) |
| Business failure | 40 (50%) |
| Lost job | 22 (28%) |
| Lost job > 6 mo | 15 (19%) |
| Child separation > 3 mo | 17 (21%) |

HAZ: Height-for-age Z-score, WAZ: Weight-for-age Z-score, EPDS: Edinburgh postnatal depression score [52], HFIAS: household food insecurity assessment scale [63], HDDS: Household Dietary diversity scale [49], HWISE: household water insecurity experience scale) [50].

cognitive function is derived from multiple different domains, a combined score was not constructed.

Thirdly, we explored the association between each growth variable (exposures) and the standardised physical function total score (outcome) using least squares regression. In addition, we explored the association between HAZ or WAZ and each individual physical function test (S9 Table and S3 Fig in S1 File). Fourthly, we explored the association between growth, years of schooling, or the child's perceived socioemotional support score (exposures) and each cognitive outcome (MPI, school achievement test, SDQ, and fine motor score); S10 and S11 Tables and S4 and S5 Figs in S1 File.

All analyses used Stata Version 15 (StataCorp LLC, College Station, TX).

## 6. Ethics

Caregivers gave written informed consent and children gave written assent to participate. Ethical approval was provided by the Medical Research Council of Zimbabwe.

## Results

### 1. SAHARAN toolbox development, feasibility and acceptability

The first objective was to test the acceptability and feasibility of the SAHARAN toolbox during the pilot measurements. The tools used showed high face validity among data collectors, district health staff, caregivers, and local community members. Body composition, anthropometry and physical function measurements were well tolerated by all children. Initial blood pressure values measured electronically were systematically high, so manual sphygmomanometers were introduced for the last 18 children. Blood pressure measurements were not included in analyses due to paucity of data. Heart rate measurement was piloted using a Fitbit Charge3 (Fitbit, USA), although the strap was too large, and subsequently altered, so this was also not included in analyses.

Children engaged well with all the cognition tests. The KABC-II required some modifications for a rural population in 2 out of the 8 subtests, as described elsewhere [61]. The School Achievement Test (SAT) was performed in 73 (91%) children (see S1 File); in the first 7 children a Shona version of the Early Grade Reading Assessment (EGRA) was performed, but scores on this tool were extremely low due to a floor effect, since the tool was too advanced for the participants tested. For 73 children in the pilot cohort, the SAT was adapted by selecting the appropriate font and choice of letters for reading that were common in Shona and Ndebele, as well as the addition of reading syllables (which are an intermediary step in reading Shona). For the pilot cohort, finger tapping was found to provide sufficient variability in test times. Rapid repeated forearm pronation and supination was also attempted but found to be uncomfortable so was discontinued. Following these modifications, all tests within the SAHARAN toolbox were feasible for data collectors and acceptable for caregivers and children.

### 2. Growth and function in the pilot cohort using the SAHARAN toolbox

Of 180 children identified by community health workers, 23 were excluded because they were outside the study age range, leaving 157 eligible children, from whom 80 were randomly selected. Among these 80 children, 3 children had the wrong age on documentation (birth certificate or health card) and were excluded, and 2 families were not interested in joining; 5 random replacements were therefore identified. Overall, 80 children (39 girls; 49%) were enrolled and underwent assessments between September 3rd and December 4th 2020.

Characteristics of this cross-sectional pilot cohort are shown in Table 1. Children had a mean age of 7.6 years (SD 0.2). Almost all households (79/80; 99%) undertook subsistence farming. Children had a mean of 3.1 years (SD 0.7) of prior schooling, with girls having more previous school exposure than boys. The impact of COVID-19 was evident: only 15/80 (19%) children had enrolled in school during that academic year, and all caregivers reported their child missing school due to COVID-19 restrictions [62]. All children completed the full battery of tests and questionnaires during a single visit, which lasted 4–5 hours.

Of note, the adversity questionnaire documented a high prevalence of adversities over the 7 years since the child's birth (Table 1), with a mean of 5 (SD 3.1) events; only one household reported no adversities and three reported only one adversity. The most common adversities were crop failure (N = 60; 75%), business failure (N = 40; 50%), lost family possessions due to hardship (N = 27; 34%), household death (N = 26; 33%), job loss (N = 22; 28%), at least one household member with an alcohol problem (N = 16; 20%), worries due to debt (N = 16; 20%) and more than 6 months of unemployment (N = 15; 19%). Of the 80 children, 17 (21%) had been admitted to hospital; only 3 children (4%) had been admitted to hospital twice or more.

**2.1 Growth and body composition.**   Mean (SD) height-for-age and weight-for-age Z-scores were -0.63 (0.81) and -0.55 (0.85), respectively, and were similar by sex; 2/80 (3%) children were stunted and 3/80 (4%) were underweight. The mean body mass index (BMI) was 15.3 kg/m$^2$ (SD 1.4, range 12.5 to 21.6). Skinfold thickness was associated with increasing HAZ and WAZ but not LMI (Fig 2A–2C). Lean mass index (LMI) was associated with increasing WAZ (Fig 2D) but not HAZ (Fig 2E). Within the child age range (7–8 years), older children did not have significantly increased growth or body composition measures.

**2.2 Physical function.**   The mean handgrip strength was 12 Kg (SD 2), and mean distance jumped in the broad jump was 111 cm (SD 16). On the shuttle run test, the average level reached was 3.4 (SD 1.2). Individual tests were associated with each other, demonstrating internal consistency, except for broad jump and shuttle-run test (S7 Table in S1 File). Children had a mean haemoglobin of 126 g/L (SD 10.1); 5 children (6%) had measurements below the WHO anaemia threshold of 110 g/L (lowest value 96 g/L), but no child had symptomatic anaemia. Older children (measured by linear regression against age) performed better at the shuttle run test (1.4 levels, 95% CI 0.3, 2.5, p = 0.01), but this was not significant for grip strength or broad jump.

**2.3 Cognition results.**   The mean scaled Mental Processing Index (MPI), which provides a total cognition score from the KABC-II, was 47 (SD 9) marks, and mean SAT score was 41 (SD 21) marks. Generally, the SAT provided a good range of variability; 18 (23%) children were unable to recognise any letters and 9 (11%) were unable to write any letters. For finger tapping, the mean fastest time to complete the task across both hands was 22 (SD 6) seconds. The mean total difficulties score for the Strengths and Difficulties Questionnaire score was 10 (SD 5) points.

Direct measurements of cognition showed internal consistency (Fig 3A–3C and S8 Table in S1 File). The MPI was strongly associated with the SAT (Fig 3A) and weakly associated with fine motor skills (Fig 3b). However, socioemotional function as measured by the caregiver-reported SDQ was not associated with SAT, MPI (Fig 3C) or fine motor function (S8 Table in S1 File). Across the 7–8 year age range of the cohort, linear regression by age showed that older children had more schooling exposure (1.6 years, 95% CI 1.0, 2.2 years, p<0.001) and performed better at the SAT (30 marks, 95% CI 12, 49 marks, p<0.001), but not for other cognitive tests.

**2.4 Caregiver questionnaire.**   The caregiver questionnaire was predominantly asked to the mother (54 out of 80, Table 1), but other primary caregivers included the father, aunt or grandmother or other carers (eg orphanage matron for one child). No child had serious

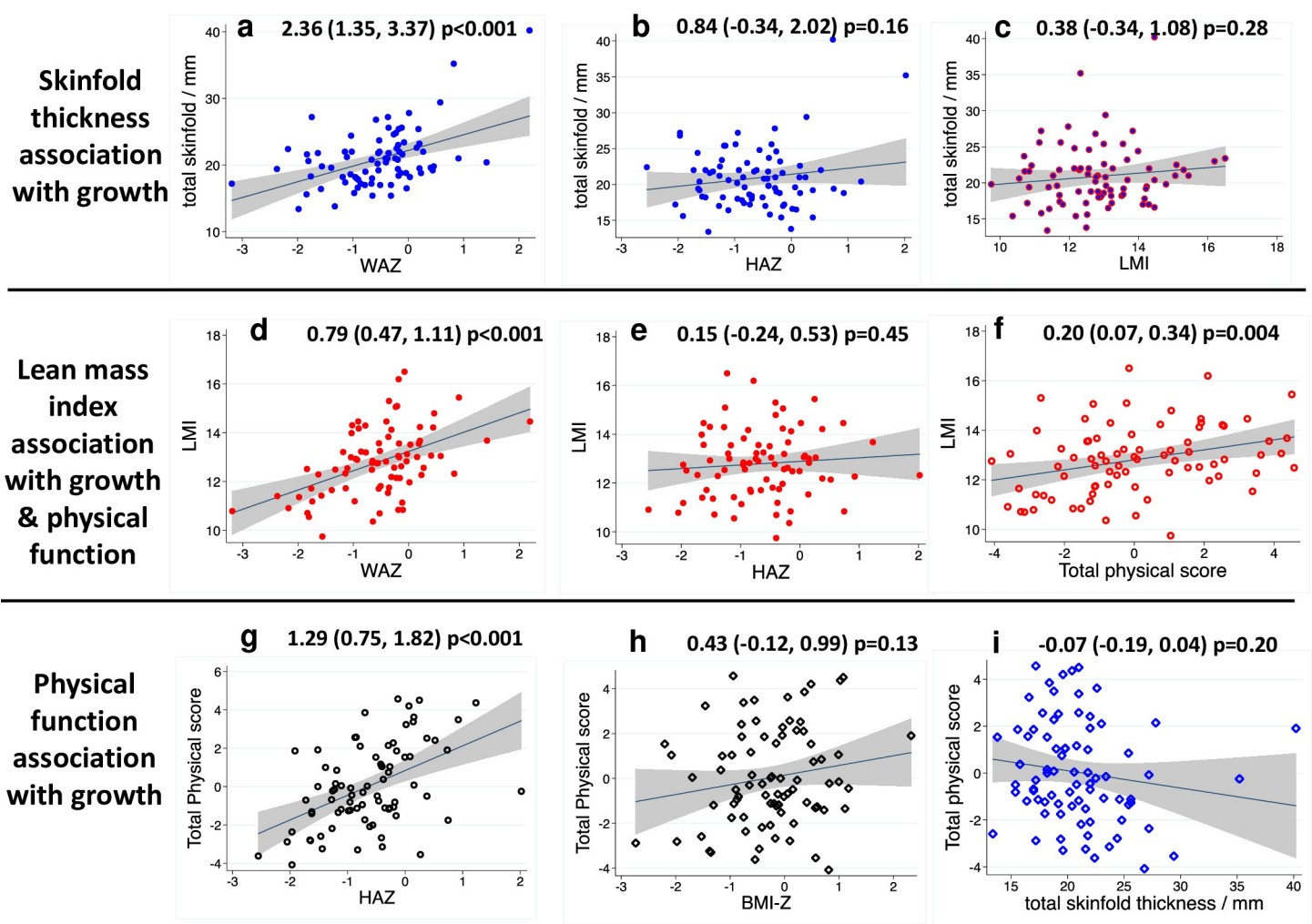

**Fig 2. SAHARAN toolbox results describing growth, body composition and their association with physical function.** (a) Total skinfold thickness was strongly associated with WAZ since fat mass increases with weight. (b & c): Total skinfold thickness was not associated with HAZ or LMI. (d): Lean mass index (LMI) was highly associated with weight-for-age Z-score (WAZ), due to increasing lean mass with weight. (e): LMI was not associated with height-for-age Z-score (HAZ) as LMI adjusts for the contribution of height to lean mass. (f): Total physical score (TPS) was strongly associated with LMI showing the positive contribution of lean mass to physical function independent of height. (g): Total physical function score (TPS) was highly associated with increasing HAZ, as lean mass increases with height. (h): TPS was not associated with Body Mass Index Z-score (BMI-Z) because of differing contributions from both fat and lean mass.(i): TPS was not associated with skinfold thickness, with a possible trend suggesting skinfold thickness may negatively contribute to total physical function.

disability. Three children (4%) had caregiver-reported problems in learning, but this was not associated with cognitive performance.

## 3. Associations between child functional domains

All physical tests were highly associated with growth measurements that included lean mass (S9 Table in S1 File and Fig 2). For every unit rise in HAZ (~6cm [56]) or WAZ (~3.5Kg [56]), maximum grip strength, broad jump, and shuttle-run test level all increased significantly. The total physical score was significantly associated with LMI (Fig 2G), HAZ (Fig 2H), WAZ, phase angle, MUAC, and calf circumference (S9 Table in S1 File). Total physical score was only weakly associated with BMI Z-score (BMI-Z) because of differing contributions from both fat and lean mass (Fig 2G). Measures of fat mass such as skinfold thickness (Fig 2H) and waist circumference (S9 Table in S1 File), were not associated with physical function. By

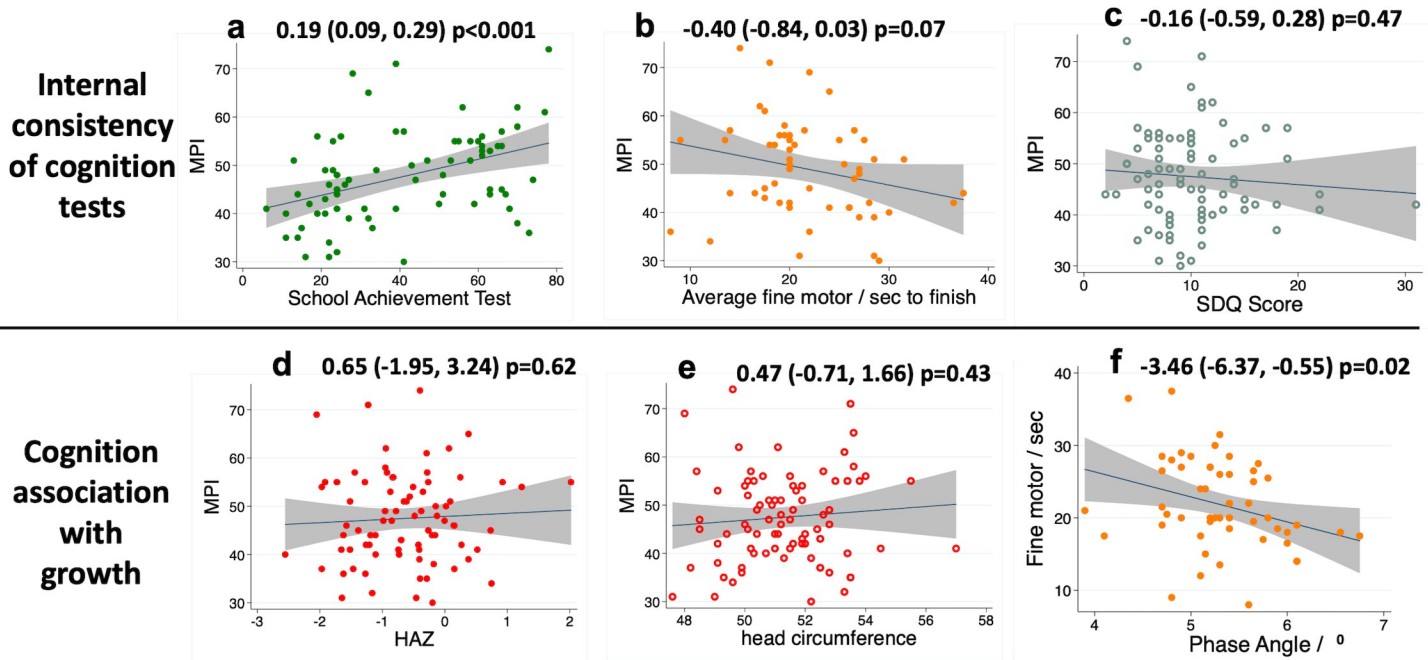

**Fig 3. SAHARAN toolbox results describing cognition results and their association with growth and physical function.** (a-c): Internal consistency showed MPI (Mental processing index) was associated with increasing School achievement test (SAT) and faster fine motor completion time but not with strength and difficulties (SDQ) score. (d & e): MPI was not associated with growth parameters of HAZ or head circumference. (f): Faster fine motor function was associated with increased bio-impedance phase angle (a marker of cellular health and membrane quality).

contrast, lean mass index (Fig 2F) and phase angle were highly associated with physical function (S9 Table in S1 File). Haemoglobin was not associated with any functional score.

By contrast, when exploring growth as an exposure and cognitive function as an outcome, there were few associations (S11 Table in S1 File). There was no association between cognition and either HAZ (Fig 3D) or head circumference (Fig 3E). Of note, a higher phase angle was significantly associated with a shorter finger tapping completion time (representing faster fine motor function) (Fig 3F), and higher MUAC was associated with a higher SAT. There was strong evidence that each additional year of schooling was associated with the SAT, and weak evidence for a relationship with the MPI; there was no association with fine motor speed or SDQ score (S10 Table in S1 File). The child's perceived socioemotional support was strongly associated with the SAT, and there was weak evidence for a relationship with the MPI score; there was no relationship between the child-reported socioemotional score and the caregiver-reported SDQ (S10 Table in S1 File).

Taken together (Fig 4), the SAHARAN toolbox showed significant associations between individual growth measures, which were strongly related to physical function. Cognitive function measurements showed internal consistency and significant associations with schooling exposure and child socioemotional score. Growth was not associated with cognitive function, except for phase angle which was associated with fine motor function.

## Discussion

There is increasing recognition of the need to assess child health, growth and function together at school-age, to more holistically evaluate long-term outcomes following early-life adversities and interventions. We formulated a conceptual framework then developed and deployed the SAHARAN toolbox to investigate the relationships between these domains in a cross-sectional

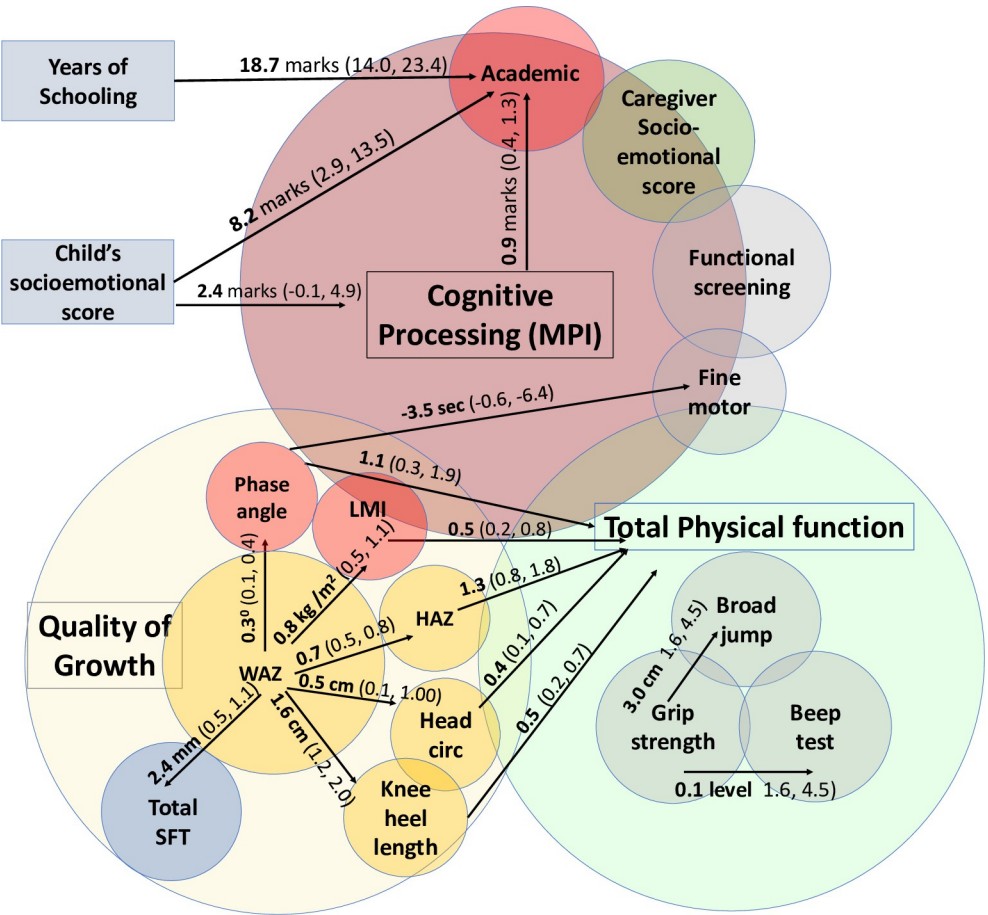

**Fig 4. Significant associations within the SAHARAN toolbox between quality of growth, total physical function and cognitive function domains.** Linear regression coefficients and confidence intervals are shown.

cohort of school-age Zimbabwean children. The tests showed good face validity and only minor modifications were needed to make them suitable for a rural 7-year-old cohort. Tests were well tolerated by both children and caregivers with appropriate levels of engagement. We were able to show internal consistency across growth, physical function and cognitive function domains. We found clear associations between linear growth and physical function across a range of tests evaluating muscle strength and cardiovascular fitness, highlighting the effect of attained height on physical performance. The quality of growth in early life was also associated with school-age measures, since lean mass index was related to physical function, independent of height. We found associations between different domains of directly measured school-age cognitive function, including global cognition and academic skills. This highlights the interconnected aspects of school-age neurodevelopment, with the potential for interventions to influence multiple cognitive domains. Collectively, our data demonstrate the feasibility and utility of combining growth, body composition, physical and cognitive function to more comprehensively unravel how early-life conditions shape prioritization and trade-offs in later function. Our novel, holistic assessment battery can now be applied to larger cohorts to gain a richer understanding of school-age health outcomes in LMIC settings.

We found significant associations within and between growth and physical function measures (summarised in Fig 4). Child height was strongly associated with all physical function

tests. Height has previously been strongly correlated with grip strength at school-age [64] as well as with strength of other muscles [65]. The strong correlations between different muscle function tests indicates a global effect of stature (possibly also mediated by bone growth) on whole-body muscle strength [66]. Height is closely associated with weight, but the association of weight with the shuttle-run test was weaker than for height, since the test measures cardio-vascular fitness. Allometric approaches that scale size with physical function typically rely on body mass and height to produce regression equations [67]. However, neither BMI nor waist circumference were associated with physical function, despite these measurements generally being recommended for body composition in adolescents and adults [15]. This is probably because of the narrow BMI range in our pilot cohort at 7 years of age. Beyond allometry, more detailed body composition measurements provided further insight into the proportion of fat and lean mass, which were closely correlated with function. MUAC, hip circumference and calf circumference were associated with physical function, as they all measure muscle as well as subcutaneous fat. When lean and fat mass were measured individually by more advanced body composition measurements, increasing lean mass index was highly associated with total physical function. This is expected as lean mass increases with both strength and fitness [68]. By contrast, fat mass represented by skinfold thickness was not significantly associated with physical function. Fat mass has previously been shown to have no relationship to grip strength [64], supporting our findings. Indeed, muscular strength is inversely associated with increasing adiposity in children and adolescents [69]. Taken together, these data provide intriguing evidence that promoting early-life linear growth and lean mass accretion may help to improve later physical function and reduce the risk of chronic disease [69, 70], supporting trends observed globally [9].

Cognitive tests demonstrated expected associations with each other, and with schooling exposure, which was impacted by the COVID-19 pandemic. The SAT score was significantly associated with years of schooling, while the mental processing index, derived from the KAB-C-II tests, had a less strong association with prior schooling; this is perhaps unsurprising, since the KABC-II is designed to include novel tests that children would not be exposed to in schools [40]. The trend towards improved fine motor skills with higher MPI is also consistent with the correlation between different domains of cognition. The fine motor tool we used has not previously been directly compared with cognition measures, although stunted compared to non-stunted Jamaican children were reported to be approximately 3 seconds slower on sequential finger tapping [41].

Each additional point on the child's perceived socioemotional score (representing the child reporting feeling happier and more supported at home) was associated with additional marks on the SAT and the MPI (Fig 4). This suggests that the child's perception of their home environment can influence their academic performance, providing some evidence of plausible future areas for psychological support. The importance of the child's perception of support has been noted when investigating risk and resilience factors for mental health [43]. By comparison, there was no association between the Strengths and Difficulties Questionnaire (SDQ) and the SAT. This may partly be because SDQ is a caregiver-reported tool, although caregiver-reported and teacher-reported SDQ were associated with academic scores among children in Brazil [71]. A recent follow-up of the MAL-ED cohort showed that socioeconomic status was associated with cognitive scores at 5 years [72]; our study also found a trend towards increasing MPI with a rise in socioeconomic scale.

There were few relationships between growth and cognitive function, except for an association between increased phase angle and improved fine motor function (Fig 3F and S11 Table in S1 File). This is plausible since higher values of phase angle are thought to reflect improved cellular health, particularly membrane function [73]. A recent systematic review

highlighted that growth is not a reliable proxy for cognitive function [13], which emphasises the importance of direct measures of cognition at school age. Using an allometric approach in larger samples, head circumference has been related to intracranial volume [74], but in low-resource environments, it may not be associated with cognitive function [75]. There is some evidence that acute exercise before cognitive testing can improve cognitive performance [76], hence we measured cognitive function routinely before physical function. Two systematic reviews have suggested a positive effect of physical activity on cognitive function but with variable and inconsistent evidence [77, 78]. Subsequent studies have shown moderate correlation in direct measurement [79], whilst two randomised trials of an exercise intervention have shown no effect on cognitive performance [80, 81].

We designed the SAHARAN toolbox for use in mid-childhood and have shown here that it is feasible to combine growth, physical and cognitive function measures. This holistic approach would be valuable across a wide range of school ages. Anthropometry measurements are routinely used in all ages, and body composition techniques can be adapted for younger ages. However, functional measurements may need adaptation. The KABC-II has been used in Africa from age 5 years [82, 83] to 16 years [84], and the SDQ from ages 3–19 years [42], although both may need piloting and adaptation for different contexts. However, measurements of literacy and numeracy are highly context- and age-specific. For example, in rural Zimbabwe, the SAT was highly dependent on previous schooling exposure and age. Finger tapping has been successfully performed in older children [41] but not to our knowledge in younger children, and it is plausible they may struggle to perform it correctly. All the physical function tests can be successfully performed in older children and adults, and the dynamometer can be adjusted for different hand sizes. There is some evidence in high-resource settings for their use in 3–5 year-olds in the PREFIT battery [85], although further piloting and adaptation may be required in low-resource settings.

The age range of this pilot cohort was narrow at 7–8 years, hence older children were not significantly taller or heavier. In this pilot cohort, there were minimal observable effects of age or biological maturity, although older children achieved a higher level in the shuttle run test, had more schooling exposure, and scored higher on the SAT compared to younger children. In general, no significant effects of age were seen on most cognitive or physical function tests. Deploying the SAHARAN toolbox on a sample with a greater age range would enable the effects of biological maturity on function to be investigated. However, cognitive tests would need to be adjusted for both schooling and effects of age with appropriate scaling, such as is available in the KABC-II. In addition, biological maturity in children in low-resource settings requires further research; techniques such as the percentage of adult height obtained have been used in high-resource settings [86], but these need to be adapted for settings with multiple adversities including a high prevalence of stunting. Nevertheless, exciting opportunities exist to combine the SAHARAN toolbox tests with physiological measures including biological maturity.

Our study has several strengths. Firstly, the SAHARAN toolbox is the first community-based, school-age assessment battery to combine measurements of school-age growth, body composition, physical and cognitive function together with a questionnaire on contemporaneous environmental factors. We included several novel tools, including sequential fine motor tapping and the use of electronic data collection in a rural setting. The toolbox is portable, and was successfully deployed during the COVID-19 pandemic. Our study also had several limitations. First, we enrolled a convenience sample of children for this study, and our findings may not be applicable to the most hard-to-reach children in all communities. However, we enrolled children from both urban and rural areas, where poverty and food insecurity are both pervasive, meaning our results may be applicable to similar settings. We performed multiple

exploratory analyses on our pilot cohort, and so individual associations may be prone to type 1 error, although associations were consistent and biologically plausible. The 4-hour test battery limits the number of children who can be measured and is a substantial time burden for families. However, future factor analyses will be undertaken to shorten the SAHARAN toolbox into the most discriminatory cognitive and non-cognitive metrics for easier deployment. The KABC-II does require considerable training, though in this study it was conducted online by experienced trainers. With permission, we adapted two KABC-II subtests which used Western images and concepts unfamiliar to rural African children [61]. Finally, this cross-sectional study only measured a single time-point, so any associations with improvements or declines in function over time could not be detected. For example, a higher adversity score may have a much greater impact in early life, which may not be detected in this cross-sectional study.

## Conclusion

We designed a holistic toolbox to fill a much-needed gap during the 'missing middle' of childhood when child health outcomes are generally overlooked. This assessment battery provides a novel combination of growth, body composition, cognitive and physical function measurements for school-age children, which would be applicable in research trials and programmes globally. Results from this study show consistency between the different cognitive measures used, and associations between growth, body composition and physical function. Our findings highlight the importance of early-life growth and relative lean mass for improving physical function and potentially reducing long-term chronic disease risk [70]. This study reaffirms the value of combining assessments of body composition, physical and cognition function, and provides the opportunity to now characterise the effects of early-life exposures and interventions on school-age growth and development in multiple settings.

## Supporting information

**S1 File.** This provides further detail on the background of the measurement tools (S1 Table), cognitive tests (S2 Table), body composition (S3 Table), physical function tests (S4 Table) and caregiver questionnaire (S5 Table). Correlation coefficients between growth variable are also presented (S6 Table) and associations between the individual physical function tests (S7 Table) and between the individual cognitive function tests (S8 Table). Exploratory analysis between growth and physical function tests are shown (S9 Table), and also the exposures of schooling years and child socioemotional score on cognitive function tests (S10 Table). Exploratory associations between growth variables and cognitive function tests are presented (S11 Table). S1 Fig demonstrates the application of the SAHARAN Toolbox in photographs. S2 Fig portrays the associations explored between individual physical function tests and between individual cognitive function tests. S3 Fig shows a diagram exploring the associations of key growth variables on total physical function, and also HAZ and WAZ growth exposures on each individual physical function test. S4 Fig presents a diagram exploring the associations between exposures of years of schooling and child's socioemotional support on each individual cognitive function test. S5 Fig demonstrates the associations between key growth variables as exposures on cognitive function tests as an outcome.
(DOCX)

## Acknowledgments

We thank Zvitambo staff for their help with this study, particularly Virginia Sauramba and Sandisiwe Ndlovu (compliance), Stephen Moyo and Peter Mapuranga (logistics), Theo Chidawanyika (IT), and Bernard Chasekwa for statistical assistance with the socioeconomic scale. We thank Prof Jean Humphrey for valuable discussions in planning this study. We thank Jackie Namukooli and Mary Nyakato for initial KABC-II training and Prof Tamsen Rochat, Samu Dube and her team for further online training for the KABC-II. We thank Prof Susan Chang-Lopez for assistance with sequential finger tapping. We also thank Dr Marko Kerac and Dr Keith Brazendale for helpful discussions regarding the physical function tests, and Dr Natasha Lelijveld and Dr Carlos Eternod-Grijalva for general advice on the SAHARAN design.

## Author Contributions

**Conceptualization:** Joe D. Piper, Clever Mazhanga, Dzivaidzo Chidhanguro, Melissa J. Gladstone, Jonathan C. Wells, Lisa F. Langhaug, Naume V. Tavengwa, Robert Ntozini, Andrew J. Prendergast.

**Data curation:** Joe D. Piper, Clever Mazhanga, Gloria Mapako, Idah Mapurisa, Tsitsi Mashedze, Eunice Munyama, Marian Mwapaura, Batsirai Mutasa, Robert Ntozini, Andrew J. Prendergast.

**Formal analysis:** Joe D. Piper, Jonathan C. Wells, Robert Ntozini, Andrew J. Prendergast.

**Funding acquisition:** Joe D. Piper, Robert Ntozini, Andrew J. Prendergast.

**Investigation:** Joe D. Piper, Clever Mazhanga.

**Methodology:** Joe D. Piper, Clever Mazhanga, Gloria Mapako, Idah Mapurisa, Tsitsi Mashedze, Eunice Munyama, Marian Mwapaura, Dzivaidzo Chidhanguro.

**Project administration:** Joe D. Piper, Clever Mazhanga, Marian Mwapaura, Dzivaidzo Chidhanguro, Lisa F. Langhaug, Naume V. Tavengwa, Robert Ntozini, Andrew J. Prendergast.

**Software:** Marian Mwapaura, Eddington Mpofu, Batsirai Mutasa, Robert Ntozini.

**Supervision:** Joe D. Piper, Clever Mazhanga, Dzivaidzo Chidhanguro, Batsirai Mutasa, Lisa F. Langhaug, Robert Ntozini, Andrew J. Prendergast.

**Validation:** Joe D. Piper, Clever Mazhanga.

**Visualization:** Joe D. Piper, Robert Ntozini, Andrew J. Prendergast.

**Writing – original draft:** Joe D. Piper, Andrew J. Prendergast.

**Writing – review & editing:** Joe D. Piper, Clever Mazhanga, Melissa J. Gladstone, Jonathan C. Wells, Robert Ntozini, Andrew J. Prendergast.

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
