## [Decision Letter · Decision Letter 0]

6 May 2022

PONE-D-21-32266

Characterising school-age health and function in rural Zimbabwe using the SAHARAN toolbox

PLOS ONE

Dear Dr. Piper,

Thank you for submitting your manuscript to PLOS ONE. After careful consideration, we feel that it has merit but does not fully meet PLOS ONE’s publication criteria as it currently stands. Therefore, we invite you to submit a revised version of the manuscript that addresses the points raised during the review process.

I would like to highlight that the submitted paper needs a lot of improvements, although one of the reviewers has added fewer issues. Please, consider this as a new opportunity to achieve the minimum standards. Unfortunately, in case the authors do not amend the major issues, the paper will be rejected in the following round. 

We look forward to receiving your revised manuscript.

Kind regards,

Javier Brazo-Sayavera, Ph.D.

Academic Editor

PLOS ONE

“Wellcome Trust provided funding for this work and the salary of JDP. Grant number: 220671/Z/20/Z "Effect of early-life nutrition and WASH interventions on the long-term health of Zimbabwean children". AJP is funded by Wellcome Trust grant number 108065/Z/15/Z. Additional funding is from National Institutes of Health (NIH) Grant number R61HD103101, and research grants from the Thrasher Research Fund and Innovative Methods and Metrics for Agriculture and Nutrition Actions (IMMANA). The funders had no role in study design, data collection and analysis, decision to publish, or preparation of the manuscript.”

“Wellcome Trust provided funding for this work and the salary of JDP. Grant number: 220671/Z/20/Z "Effect of early-life nutrition and WASH interventions on the long-term health of Zimbabwean children". AJP is funded by Wellcome Trust grant number 108065/Z/15/Z. Additional funding is from National Institutes of Health (NIH) Grant number R61HD103101, and research grants from the Thrasher Research Fund and Innovative Methods and Metrics for Agriculture and Nutrition Actions (IMMANA). The funders had no role in study design, data collection and analysis, decision to publish, or preparation of the manuscript.”

5. Please note that in order to use the direct billing option the corresponding author must be affiliated with the chosen institute. Please either amend your manuscript to change the affiliation or corresponding author, or email us at plosone@plos.org with a request to remove this option.

Reviewers' comments:

Reviewer's Responses to Questions

**Comments to the Author**

1. Is the manuscript technically sound, and do the data support the conclusions?

Reviewer #1: Partly

Reviewer #2: Yes

2. Has the statistical analysis been performed appropriately and rigorously? 

Reviewer #1: No

Reviewer #2: Yes

3. Have the authors made all data underlying the findings in their manuscript fully available?

Reviewer #1: Yes

Reviewer #2: Yes

4. Is the manuscript presented in an intelligible fashion and written in standard English?

Reviewer #1: Yes

Reviewer #2: Yes

5. Review Comments to the Author

Reviewer #1: The manuscript brings relevant data and results; however, the paper has a lot of information, making it difficult for the reader to understand what gaps the study aims at closing.

Writing and format editing is required.

Based on the abstract, it seems that the authors are divided between two approaches: a) to describe the SAHARAN toolbox development, or b) to verify the association between nutritional status, physical and cognitive function. The first major change that needs to be addressed is to adequately define which are the aims of the present study and to focus the description of the entire manuscript accordingly.

In my understanding the data presented here is the baseline, is that correct? Please provide further description of the cohort.

Statistical analysis is also a concerning topic. The paper needs a clear outline prior to data analysis. The authors seem to have analyzed all domains available from the toolbox without defining which variable is considered as outcome, exposure, mediator, or adjusting variable. As a consequence, Figures 2 and 3 have different outcome variables, making it very confusing for the reader.

A few descriptive figures/tables, only illustrating the variables, would be interesting.

Table 1 needs formatting revision to be fit for publication.

Reviewer #2: Reviewer’s comment

Abstract: try to reduce abbreviations in the abstract section

Line 24-27: The introduction needs to re-write clearly and attract the readers.

Line 28-36: The ‘SAHARAN toolbox’ seems more technical and should be explained more for all readers.

Line 39: The number isn’t written in the figure at the beginning of sentences. Instead, you can write it in full like ‘Eighty’.

Line 124: Do you have justification why you’ve started the methods by Conceptual framework? Could you explain how you’ve used the framework, borrowed or adopted from previous work.

Line 334: When you say’ caregiver questionnaire, is that not include parents?

6. PLOS authors have the option to publish the peer review history of their article (what does this mean?). If published, this will include your full peer review and any attached files.

Reviewer #1: No

Reviewer #2: No

---

## [Author Response · Author response to Decision Letter 0]

15 Nov 2022

Thank you for the detailed reviews and feedback. Please find our revised manuscript and supplementary information attached. Please note that we have made substantial changes to the manuscript in response to the reviewers’ feedback. These are described in detail in the “Response to reviewers” attached document. Overall, we have updated the formatting in line with PLOS recommendations. We have completed the inclusivity in Global Health Research questionnaire. We have provided an updated funding statement below which was requested to be included in the cover letter.

We have uploaded the minimum de-identified dataset and supporting materials, which are available on Open Science Framework at https://osf.io/4m6zs/. The author Joe D Piper remains affiliated with Queen Mary University of London and this University does allow direct billing: please contact Joe Piper directly if there are any further issues on j.piper@qmul.ac.uk

In response to the reviewer’s comments, we have clarified throughout the manuscript the dual aims of developing the SAHARAN toolbox and then its use in a pilot cohort of 80 school-age Zimbabwean children. We have also clarified throughout the manuscript that this was a pilot cross-sectional cohort study of 80 children. We have considerably updated the data analysis section in both the main text and supplementary information. We have improved the clarity of defining the exposures and outcomes before presenting the results. We have updated both figure 2 and figure 3. We have also added 3 further supplementary figures to provide further clarity on the analysis, including variables, exposures and outcomes. We have provided additional description for the SAHARAN toolbox and explained that we used the conceptual framework to select the tests and structure the toolbox. We have also rewritten the introduction and clarified the caregiver questionnaire was primarily asked to mothers, or whoever the alternative caregiver was when the mother was not available. 

We look forward to receiving your feedback.

---

## [Decision Letter · Decision Letter 1]

23 Jan 2023

PONE-D-21-32266R1Characterising school-age health and function in rural Zimbabwe using the SAHARAN toolboxPLOS ONE

Dear Dr. Piper,

Thank you for submitting your manuscript to PLOS ONE. After careful consideration, we feel that it has merit but does not fully meet PLOS ONE’s publication criteria as it currently stands. Therefore, we invite you to submit a revised version of the manuscript that addresses the points raised during the review process.

It has been difficult to recruit the former reviewers and one has been substitued. However, this has taken more time than expected because several reviewers refused the invitation to review the submission. I apologise for the delay. At this point, I consider relevant to address the issuess mentioned by the reviewers. Although reviewer 1 has accepted, a couple of comments should be addressed. 

We look forward to receiving your revised manuscript.

Kind regards,

Javier Brazo-Sayavera, Ph.D.

Academic Editor

PLOS ONE

Journal Requirements:

Reviewers' comments:

Reviewer's Responses to Questions

**Comments to the Author**

1. If the authors have adequately addressed your comments raised in a previous round of review and you feel that this manuscript is now acceptable for publication, you may indicate that here to bypass the “Comments to the Author” section, enter your conflict of interest statement in the “Confidential to Editor” section, and submit your "Accept" recommendation.

Reviewer #1: All comments have been addressed

Reviewer #3: (No Response)

2. Is the manuscript technically sound, and do the data support the conclusions?

Reviewer #1: Yes

Reviewer #3: Partly

3. Has the statistical analysis been performed appropriately and rigorously? 

Reviewer #1: Yes

Reviewer #3: Yes

4. Have the authors made all data underlying the findings in their manuscript fully available?

Reviewer #1: Yes

Reviewer #3: Yes

5. Is the manuscript presented in an intelligible fashion and written in standard English?

Reviewer #1: Yes

Reviewer #3: Yes

6. Review Comments to the Author

Reviewer #1: Authors have significantly improved the manuscript, meeting virtually all the concerns raised in the previous revision. Congratulations.

I will only suggest minor changes:

1- Include the aims of the paper in the abstract

2- Table 1 needs formatting revision to be fit for publication (this comment was not addressed)

Reviewer #3: - I thank the authors for their good job in the first revision round. However, I think that the aims of the study are still confusing. The arguments used in the introduction (that I agree) is based on the need of diagnosis/surveillance of holistic health condition on LMIC children. However, the main results and discussion are on associations. So, I believe that at least three aims should be address:

1) Development, implementation and feasibility of the toolbox;

2) Descriptive characteristics of the sample (arguments used in the introduction);

3) Association between the components of the SAHARAN toolbox (further theoretical framework and scientific gaps should be described for that aim, even making it clear in the title).

I am not sure theses aims should be in the same article considering that each aim would have a specific organization based on guidelines (please see: https://www.equator-network.org/).

- For the aim 1, are there quality indicators of the tests and questionnaires used (e.g. reproducibility, familiarization, psychometrics properties, etc)?

- I do suggest authors explore allometric adjustment and to discussion the role of the biological maturation in the results found and for the use of the toolbox in the future (aims 2 and 3).

- Please provide details about the assessment conducted. For example, a) was any adaptation done in the physical tests? Even with details in the supplementary files, I do suggest to include it generally in the main document; b) it is not possible to understand about the hemodynamic measures: “additional measurements of blood pressure, heart rate and haemoglobin”; b) would the cognitive test the same for 5 and 14 years old children?

7. PLOS authors have the option to publish the peer review history of their article (what does this mean?). If published, this will include your full peer review and any attached files.

Reviewer #1: No

Reviewer #3: **Yes: **Danilo Rodrigues Pereira da Silva

---

## [Author Response · Author response to Decision Letter 1]

10 Apr 2023

Reviewers' comments:

Reviewer's Responses to Questions

Comments to the Author

Reviewer #1: Authors have significantly improved the manuscript, meeting virtually all the concerns raised in the previous revision. Congratulations.

I will only suggest minor changes:

1- Include the aims of the paper in the abstract

Response: Thank you for this: the aims are now included in the abstract

2- Table 1 needs formatting revision to be fit for publication (this comment was not addressed)

Response: Apologies that this was missed: this has now been re-formatted using the instructions to authors guidelines.

Reviewer #3: - I thank the authors for their good job in the first revision round. However, I think that the aims of the study are still confusing. The arguments used in the introduction (that I agree) is based on the need of diagnosis/surveillance of holistic health condition on LMIC children. However, the main results and discussion are on associations. So, I believe that at least three aims should be address:

1) Development, implementation and feasibility of the toolbox;

2) Descriptive characteristics of the sample (arguments used in the introduction);

3) Association between the components of the SAHARAN toolbox (further theoretical framework and scientific gaps should be described for that aim, even making it clear in the title).

Response: Thank you for this suggestion. We have now incorporated all these aims in the abstract, introduction, methods, results and discussion sections. 

I am not sure theses aims should be in the same article considering that each aim would have a specific organization based on guidelines (please see: https://www.equator-network.org/).

Response: We have further clarified that this was a pilot cross-sectional study in the methods, and hence we have emphasised that we have followed the Guidelines for reporting non-randomised pilot and feasibility studies1 (Lancaster et al. 2019, Pilot and Feasibility studies) combined with the STROBE checklist for observational epidemiological studies from equator-network2. 

We feel these aims should be in the same article since the pilot cohort reinforces the first aim of demonstrating the implementation and feasibility of the SAHARAN toolbox. We also feel that the exploratory analyses demonstrate a useful example for future investigations using larger, more detailed or randomised samples. 

- For the aim 1, are there quality indicators of the tests and questionnaires used (e.g. reproducibility, familiarization, psychometrics properties, etc)?

Response: Thank you for this question. These quality indicators form part of the COSMIN checklist in Table S1. This pilot study was not able to measure the children multiple times or provide more detailed analysis. Additional information on these has been flagged in Table S1 with definitions provided in the table footnote for validity, reliability and responsiveness. For familiarisation, further details have been added in the main paper and also the validity section of supporting information to describe any changes made to improve validity. 

- I do suggest authors explore allometric adjustment and to discussion the role of the biological maturation in the results found and for the use of the toolbox in the future (aims 2 and 3).

Response: Concepts of allometry and biological maturation have been added to the discussion section. Interestingly, the body composition methods provide additional detail beyond allometry with the associations of lean mass but not skinfold thicknesses on total physical function. Regarding biological maturation, associations with age of the cohort are also discussed by performing linear regression across the age range of the cohort (although the age range of 7-8 years is relatively narrow). Combining the toolbox with measures of biological maturity in the future is also now described in the discussion. 

- Please provide details about the assessment conducted. For example, a) was any adaptation done in the physical tests? Even with details in the supplementary files, I do suggest to include it generally in the main document; b) it is not possible to understand about the hemodynamic measures: “additional measurements of blood pressure, heart rate and haemoglobin”; b) would the cognitive test the same for 5 and 14 years old children?

Response: We appreciate this suggestion. We have now added further details in the methods section of the paper. We have also added additional details on the haemodynamic measures. The ability to use this tool and any modifications required within different ages has also been clarified with the addition of a separate paragraph in the discussion section. 

1. Lancaster GA, Thabane L. Guidelines for reporting non-randomised pilot and feasibility studies. Pilot and Feasibility Studies 2019; 5(1): 114.

2. von Elm E, Altman DG, Egger M, Pocock SJ, Gøtzsche PC, Vandenbroucke JP. The Strengthening the Reporting of Observational Studies in Epidemiology (STROBE) statement: guidelines for reporting observational studies. J Clin Epidemiol 2008; 61(4): 344-9.

---

## [Decision Letter · Decision Letter 2]

19 Apr 2023

PONE-D-21-32266R2Characterising school-age health and function in rural Zimbabwe using the SAHARAN toolboxPLOS ONE

Dear Dr. Piper,

Thank you for submitting your manuscript to PLOS ONE. After careful consideration, we feel that it has merit but does not fully meet PLOS ONE’s publication criteria as it currently stands. Therefore, we invite you to submit a revised version of the manuscript that addresses the points raised during the review process.

 Although the reviewer 3 considers that the authors have progressed accordingly respecting the previous version, there are still some minor issues to address. I encourage the authors to amend them following reviewer's suggestion. 

We look forward to receiving your revised manuscript.

Kind regards,

Javier Brazo-Sayavera, Ph.D.

Academic Editor

PLOS ONE

Journal Requirements:

Reviewers' comments:

Reviewer's Responses to Questions

**Comments to the Author**

1. If the authors have adequately addressed your comments raised in a previous round of review and you feel that this manuscript is now acceptable for publication, you may indicate that here to bypass the “Comments to the Author” section, enter your conflict of interest statement in the “Confidential to Editor” section, and submit your "Accept" recommendation.

Reviewer #3: (No Response)

2. Is the manuscript technically sound, and do the data support the conclusions?

Reviewer #3: Yes

3. Has the statistical analysis been performed appropriately and rigorously? 

Reviewer #3: Yes

4. Have the authors made all data underlying the findings in their manuscript fully available?

Reviewer #3: (No Response)

5. Is the manuscript presented in an intelligible fashion and written in standard English?

Reviewer #3: (No Response)

6. Review Comments to the Author

Reviewer #3: I thank the authors for the good job on the paper. Among my comments in the last round of revision, I only feel that further theoretical framework and scientific gaps should be described for the third aim, even making it clear in the title. In addition, I recommend to move the aims (below) to the end of the introduction section instead of be in the "methods".

"There were three objectives to this study: first, to develop the SAHARAN toolbox as a holistic measure of school-age growth, cognitive and physical function and demonstrate its feasibility and acceptability; second, to pilot it in a cohort of children in rural Zimbabwe to describe their growth and function; and third, to explore associations between key exposures and outcomes in this pilot cohort."

7. PLOS authors have the option to publish the peer review history of their article (what does this mean?). If published, this will include your full peer review and any attached files.

Reviewer #3: **Yes: **Danilo Rodrigues Pereira da Silva

---

## [Author Response · Author response to Decision Letter 2]

24 Apr 2023

Please note our detailed response to the reviewer's minor revisions below: 

Response to reviewers

Reviewer #3: I thank the authors for the good job on the paper. Among my comments in the last round of revision, I only feel that further theoretical framework and scientific gaps should be described for the third aim, even making it clear in the title. 

Thank you for this suggestion: we have included further theoretical framework and scientific gaps for the third aim in the introduction, explaining how modern techniques of body composition, physical function and detailed cognitive function can be combined. This provides further framework for the third aim. However, we would prefer to keep the current title of the manuscript as we feel this adequately describes the study. 

In addition, I recommend to move the aims (below) to the end of the introduction section instead of be in the "methods".

"There were three objectives to this study: first, to develop the SAHARAN toolbox as a holistic measure of school-age growth, cognitive and physical function and demonstrate its feasibility and acceptability; second, to pilot it in a cohort of children in rural Zimbabwe to describe their growth and function; and third, to explore associations between key exposures and outcomes in this pilot cohort."

Thank you for this suggestion: we have moved this to the introduction section.

---

## [Decision Letter · Decision Letter 3]

27 Apr 2023

Characterising school-age health and function in rural Zimbabwe using the SAHARAN toolbox

PONE-D-21-32266R3

Dear Dr. Piper,

We’re pleased to inform you that your manuscript has been judged scientifically suitable for publication and will be formally accepted for publication once it meets all outstanding technical requirements.

Kind regards,

Javier Brazo-Sayavera, Ph.D.

Academic Editor

PLOS ONE

Reviewer's Responses to Questions

**Comments to the Author**

1. If the authors have adequately addressed your comments raised in a previous round of review and you feel that this manuscript is now acceptable for publication, you may indicate that here to bypass the “Comments to the Author” section, enter your conflict of interest statement in the “Confidential to Editor” section, and submit your "Accept" recommendation.

Reviewer #3: (No Response)

2. Is the manuscript technically sound, and do the data support the conclusions?

Reviewer #3: (No Response)

3. Has the statistical analysis been performed appropriately and rigorously? 

Reviewer #3: (No Response)

4. Have the authors made all data underlying the findings in their manuscript fully available?

Reviewer #3: (No Response)

5. Is the manuscript presented in an intelligible fashion and written in standard English?

Reviewer #3: (No Response)

6. Review Comments to the Author

Reviewer #3: (No Response)

7. PLOS authors have the option to publish the peer review history of their article (what does this mean?). If published, this will include your full peer review and any attached files.

Reviewer #3: **Yes: **Danilo Rodrigues Pereira da Silva

---

## [Editor Report · Acceptance letter]

3 May 2023

PONE-D-21-32266R3 

Characterising school-age health and function in rural Zimbabwe using the SAHARAN toolbox 

Dear Dr. Piper:

I'm pleased to inform you that your manuscript has been deemed suitable for publication in PLOS ONE. Congratulations! Your manuscript is now with our production department. 

Kind regards, 

on behalf of

Dr. Javier Brazo-Sayavera 

Academic Editor

PLOS ONE